# The Impacts of Calamity Logging on the Sustainable Development of Spruce Fuel Biomass Prices and Spruce Pulp Prices in the Czech Republic

**Mansoor Maitah** [1], **Daniel Toth** [2], **Karel Malec** [1], **Seth Nana Kwame Appiah-Kubi** [1,*], **Kamil Maitah** [1], **Dariusz Pańka** [3], **Piotr Prus** [4], **Jaroslav Janků** [5] and **Robert Romanowski** [6]

1   Department of Economics, Faculty of Economics and Management, Czech University of Life Sciences in Prague, Kamýcká 129, 165 00 Prague, Czech Republic; maitah@pef.czu.cz (M.M.); maleck@pef.czu.cz (K.M.); maitahk@pef.czu.cz (K.M.)
2   Department of Economics, The University College of Business in Prague, Spálená 76/14, 110 00 Prague, Czech Republic; toth@vso-praha.eu
3   Department of Biology and Plant Protection, Faculty of Agriculture and Biotechnology, Bydgoszcz University of Science and Technology, 85-796 Bydgoszcz, Poland; dariusz.panka@pbs.edu.pl
4   Laboratory of Economics and Agribusiness Advisory, Department of Agronomy, Faculty of Agriculture and Biotechnology, Bydgoszcz University of Science and Technology, 430 Fordońska Street, 85-790 Bydgoszcz, Poland; piotr.prus@pbs.edu.pl
5   Department of Pedology, Faculty of Agrobiology, Food and Natural Resources, Czech University of Life Sciences in Prague, Kamýcká 129, 165 00 Prague, Czech Republic; janku@af.czu.cz
6   Department of Commerce and Marketing, Poznań University of Economics and Business, Al. Niepodległości 10, 61-875 Poznań, Poland; Robert.Romanowski@ue.poznan.pl
*   Correspondence: appiah-kubi@pef.czu.cz; Tel.: +420-608-399-941

**Abstract:** Currently, due to the calamity of unplanned harvesting, the amount of biomass from wood products has increased. Forests occupy 33.7% of the total area of the Czech Republic; therefore, wood and non-wood forest products are important renewables for the country. Wood biomass consists mainly of branches and bark that are not used in the wood or furniture industry. However, it can be used in bioenergy, including wood processing for fuel. As spruce production in the Czech Republic increased from the planned 15.5 million to almost 36.8 million trees in 2020, the price of wood biomass can be expected to be affected. This study aims to develop a predictive model for estimating the decline in the price of wood biomass for wood processors, such as firewood or sawdust producers, as well as for the paper industry. Wood biomass prices are falling with each additional million m$^3$ of spruce wood harvested, as is the decline in wood pulp, which is intended for the paper and packaging industries. The proposed predictive model based on linear regressions should determine how the price of wood biomass will decrease with each additional million harvested spruce trees in the Czech Republic. This tool will be used for practical use in the forestry and wood industry. The linear regression model is suitable for practical forestry use due to its simplicity and high informative value. The aim of the research is to model the dependence of the prices of firewood in the form of wood briquettes and pellets for domestic and industrial processing, as well as the prices of wood pulp on the volume of unplanned logging. It is a guide for the practice of how to use excess spruce wood from unplanned mining in the field of alternative processing with a sustainable aspect for households or heat production for households. The intention is to carry out modelling in such a way that it does not include prices of higher quality wood assortments, which are intended for the woodworking industry.

**Keywords:** biomass; forestry economy; spruce wood; timber trade; forest management

## 1. Introduction

The European Commission (EC) supports the production of renewable energy sources and biomass bioenergy in a biological development strategy that includes the production

of renewable biological resources and their conversion into food, bioenergy, and biomass products [1]. This is a long-term challenge for industries involved with agriculture, forestry, fishing, food, pulp, and paper, as well as for the chemical, biotechnology, and energy industries. These sectors should increasingly contribute to bioeconomic processes [1].

For these reasons, the promotion of biomass production is a very important part of European energy policies and environmental strategy [1]. Many of these strategies have been further developed from a socio-economical viewpoint to improve the economy and market competitiveness, including increasing the supply of jobs while at the same time working towards sustainable forest management. Winkel [2] states that forest bioeconomy is an economic activity involving forests and forest ecosystem services, including biomass-based value chains and the economic exploitation of other forest ecosystem resources energy [1,3]. Wood production also lends to the development of biomass and renewable energy sources, which are part of the traditional use of wood and timber materials in European countries [4]. In addition to wood products, forests offer valuable forestry services and functions [5]. Forests provide support services that are essential to life such as oxygen, and provide soil, and climate protection. In the Czech Republic, forests cover 2.9 million ha, which is 33.7% of the total area of land [6]. This is significantly less than the European average of 42%. In Austria it is 45%, and in Germany 31% of the land is forested. Based on the Czech Forest Report in 2018, forestry, including logging and the woodworking industry, accounted for 1.2% of gross domestic product [6]. Demand for renewable resources is currently rising and it is a political priority to replace fossil fuels with these renewable resources. Therefore, forestry is a key sector and the formulation of a long-term bioeconomic strategy is necessary. Soon, forestry will likely shift from mere wood production in poorly regulated forests to socially and environmentally sustainable forest management [7].

In Central Europe, the main wood biomass is spruce, sawdust, tree bark, and wood chips [5]. The increased and unplanned production of spruce due to the bark beetle calamity affects not only the price of spruce wood but also the price of biomaterial waste [1,3]. Unplanned calamity extraction also increases the amount of waste material in the form of bark and branches. After the basic processing of spruce wood using a shredder, the wood is processed into chips and other biomass assortments.

Bioeconomic strategy was not mentioned in the Czech National Forestry Program [8] but this plan was proposed in the draft strategy of the Ministry of Agriculture in 2018. In addition to logging, wood products and biomass are the basic priorities of the Czech forestry sector, as well as forest fruits and mushrooms [8]. The primary development of paper products involves the vertical process of pulp production. Revenue for paper companies relates to the growth of demand in the economy. The price of paper products is strongly influenced by the production capacities in Europe since there is strong competition within the paper and pulp industry. These prices are also significantly affected by the increased production of wood biomass and unplanned logging of spruce due to damages from the bark beetle. Additionally, paper consumption in the Czech Republic will increase due to the growing demand for packaging materials [9–14], as well as medical and sanitary paper materials. However, the consumption of newsprint and graphic paper is expected to decrease [11–14].

Wood for construction purposes is the main product of forestry. More specifically, these wood products are most often logs [15,16] used for industrial purposes, pulp for the production of paper and packaging material, and wood chips utilized for the production of boards and furniture [17,18]. Waste wood materials, such as sawdust, are increasingly becoming a part of industrial wood processing, as are bark chips and waste from trees such as branches [18]. Between 2000 and 2020, the Czech Statistical Office also published data on the number of wooden houses and building structures, which have increased by 15% [19]. In 2020, the Czech Statistical Office stated that about 25% of energy used comes from renewable sources and biomass. From 1995 to 2020, the trend in the use of energy from renewable sources doubled. Households use wood and wood biomass as a renewable

energy source (66.5%). More than 25.2% of bioenergy was used for energy consumption, followed by other sectors with 8.3% [17].

In Czech households, the most common source of energy is firewood, as it is very cost-effective. Studies from Finland, Germany, and Austria have reported growing economic activity in rural areas, which includes energy production from biomass [19–21]. A limiting factor for the further use of wood biomass is the shortage of workers, who are necessary for the processes of logging and wood processing. At present, the Czech forestry sector lacks more than 6000 employees [21]. In the Czech Republic, the number of employees in the forestry sector has been declining over the last two decades, mainly due to low wages. The shortage of workers, specifically in logging, has been especially evident in the last two years, when forests were massively damaged by bark beetles [21]. With limited human resources it is not possible to manage the removal of infested trees within the legal deadline. During planting, there is another acute shortage of workers and there are not enough workers to grow trees. Therefore, European countries are improving support for bioenergy and the bioeconomy in the long term [22], which should lead to a greater interest in employment in the forestry sector and its value chains in the future [23].

However, maintaining the production of wood biomass and pulp to produce paper and packaging material is important from the point of view of international trade. In this context, the European Commission has issued an action plan to promote the bioeconomy [23]. In the European Union, only 60 percent of home-produced wood biomass is processed in green energy [13,15]. The European Union also defends wood biomass on the grounds of energy security. The EU produces 96 percent of the wood biomass that is consumed there. A total of 60 percent of the biomass is wood and wood products that come from the wood processing sector. In the world, in Europe and the Czech Republic, wood production is increasing, mainly due to unplanned calamitous logging.

The aim of this research is, therefore, to propose an economic-mathematical model that can determine the future price of firewood depending on the additional increase in spruce wood production.

## 2. Materials and Methods

A predictive model was used in our methodical procedure [24]. This is a common statistical task in which the interdependence of two (or more) variables and their mathematical expression is determined. For this purpose, is used various predictive methods to find a suitable function f (x), to approximate the relationship between the measured quantities. One of the most common methods is the least squares method. Let us have n-measured pairs [$x_i$; $y_i$], which interleave the curve determined by the equation:

$$y = f(x) \tag{1}$$

We look for a function f(x) that has a minimum sum of squares y of coordinates of the measured points and points lying on the interlaced curve:

$$S = \sum_{i=1}^{n} (y_i - f(x_i))^2 \tag{2}$$

The mathematical calculation is based on partial derivatives equal to zero [10,11]. In general, this procedure can be used for many f (x) functions, but the most commonly used, line, is approximate data:

$$y = f(x) = k \cdot x + e \tag{3}$$

In our case, we will use linear regression. The exact derivation of the regression coefficients k and e is as follows:

$$k = \frac{n \cdot \left(\sum_{i=1}^{n} x_i y_i\right) - \left(\sum_{i=1}^{n} x_i\right) \cdot \left(\sum_{i=1}^{n} y_i\right)}{n \cdot \left(\sum_{i=1}^{n} x_i y_i{}^2\right) - \left(\sum_{i=1}^{n} x_i\right)^2} \tag{4}$$

$$e = \frac{\left(\sum_{i=1}^{n} x_i^2\right) \cdot \left(\sum_{i=1}^{n} y_i\right) - \left(\sum_{i=1}^{n} x_i\right) \cdot \left(\sum_{i=1}^{n} x_i y_i\right)}{n \cdot \left(\sum_{i=1}^{n} x_i^2\right) - \left(\sum_{i=1}^{n} x_i\right)^2} \tag{5}$$

The suitability of using linear regression [24–26] is justified by the correlation coefficient $r_{xy}$, whose value lies in the interval $<-1; >1$. A straight-line approximation is justified if $|r_{xy}| > 0.99$ (the two-nine rule). The following applies to the calculation:

$$r_{xy} = \frac{\sum_{i=1}^{n} (x_i - \bar{x})(y_i - \bar{y})}{\sqrt{\sum_{i=1}^{n} (x_i - \bar{x})^2 \sum_{i=1}^{n} (y_i - \bar{y})^2}} \tag{6}$$

For deviations of the found regression coefficients, the following relations apply:

$$\sigma_k = \sqrt{\frac{S_0}{(n-2) \cdot \left[\sum_{i=1}^{n} x_i^2 - \frac{1}{n} \cdot \left(\sum_{i=1}^{n} x_i\right)^2\right]}} \tag{7}$$

$$\sigma_k = \sqrt{\frac{S_0 \cdot \frac{1}{n} \cdot \sum_{i=1}^{n} x_i^2}{(n-2) \cdot \left[\sum_{i=1}^{n} x_i^2 - \frac{1}{n} \cdot \left(\sum_{i=1}^{n} x_i\right)^2\right]}} \tag{8}$$

$$S_0 = \left(\sum_{i=1}^{n} y_i^2\right) - \frac{1}{n} \cdot \left(\sum_{i=1}^{n} y_i\right)^2 - k \cdot \left[\sum_{i=1}^{n} x_i y_i - \frac{1}{n} \cdot \left(\sum_{i=1}^{n} x_i\right) \cdot \left(\sum_{i=1}^{n} y_i\right)\right] \tag{9}$$

The confidence interval is determined together with the calculation of the regression coefficients. Their accuracy depends on the variations and the calculation of the probability P. The coefficient tP, $(n-1)$ has the parameters $n-1$ and $p = 95\%$ [26,27]. Random extraction is indicated in our calculation as an independent variable (X). The dependent variable (Y) will refer to the average price of spruce biomass. $M_x$ is the average. $SS_x$ is the total sum of squares. The sum of squares ($SS_x$) used to measure the degree of variability of a single variable is defined as:

$$SS_x = \sum_{i=1}^{n} (X - M_x)^2 \tag{10}$$

$SP_{xy}$ is the sum of products that is used to measure the variability shared between two variables. It is defined as:

$$SP_{xy} = \sum_{i=1}^{n} (X - M_x)(Y - M_y) \tag{11}$$

$SP_{xy}$ is an abbreviation for the sum of products corresponding to the scattering score for two variables. To calculate SPX, we first determine the deviation score for each X and Y. $SP_{xy}$ can be calculated for each pair of deviation scores and can add them. Next, a regression line can be created that expresses the linear dependence between the defined variables. The formation of the regression line is based on a general mathematical notation [XXX]. The assumption is that the existence of an extreme function of two or more variables is zero on the first partial derivatives, in our case:

$$\frac{\partial S^2(\beta_1, \beta_2)}{\partial \beta_1} = \frac{\partial S^2(\beta_1, \beta_2)}{\partial \beta_2} = 0 \tag{12}$$

The condition sufficient for the minimum need not be investigated, since the function $S(\beta_1, \beta_2)$ is purely convex. Thus, the following applies:

$$\frac{\partial S^2(\beta_1, \beta_2)}{\partial \beta_1} = 2 \sum_{i=1}^{n} (y_i - \beta_1 - \beta_2 x_i)(-1) = 0 \tag{13}$$

$$\frac{\partial S^2(\beta_1, \beta_2)}{\partial \beta_2} = 2 \sum_{i=1}^{n}(y_i - \beta_1 - \beta_2 x_i)(-x_i) = 0 \tag{14}$$

Then, a system of normal equations can be obtained

$$\beta_1 n + \beta_2 \sum_{i=1}^{n} x_i = \sum_{i=1}^{n} y_i \tag{15}$$

$$\beta_1 \sum_{i=1}^{n} x_i + \beta_2 \sum_{i=1}^{n} y_i^2 = \sum_{i=1}^{n} x_i y_i \tag{16}$$

The system of equations is then given as a tool to create a predictive model that can be used to quantify an estimate of a dependent variable, i.e., raw spruce prices in category II. (spruce/fir). In our research, it is necessary to investigate dependencies where there may not be a clear relationship between the observed properties (random variables). In this case, can be applied a stochastic dependence [7,27]. This type of statistical dependence can be shown for both variables (variable X is the volume of timber harvesting and variable Y is the price of spruce timber in class II). As a result, a null hypothesis can be formulated:

$$y = f(x) \tag{17}$$

$$H_0: \rho_{X,Y} = 0 \tag{18}$$

That is, there is no linear relationship between X and Y. In the case of an alternative hypothesis, it makes sense to consider three options:

$$H_{A1}: \rho_{X,Y} \neq 0 \tag{19}$$

It can be applied the two-way alternative if the calculated correlation coefficient is close to 0.

$$H_{A2}: \rho_{X,Y} > 0 \tag{20}$$

It can be applied the right-sided alternative if the calculated correlation coefficient is greater than 0, so the sample indicates a positive linear dependence.

$$H_{A3}: \rho_{X,Y} < 0 \tag{21}$$

It can be applied left-sided alternative if the calculated correlation coefficient is less than 0; therefore, the sample indicates a negative linear dependence [7,27]. For verification it is necessary to calculate r:

$$r = \sum_{i=1}^{n} \left( \frac{Y - M_x}{\sqrt{(SS_x)(SS_y)}} \right) \tag{22}$$

The Pearson correlation coefficient determines whether it is a linear regression dependence, a right alternative, or a left alternative. Only then can the hypothesis be evaluated (to reject zero and accept one of the alternative hypotheses). These statistical methods correspond to economic theory, which states that the quantity of goods affects its price. It is predictable how the price of wood biomass will change with an increase of one million spruce timber harvested in an unplanned logging regime caused by biological factors, specifically damages from bark beetle. Data for calculations are predominantly obtained from the database of the Czech Statistical Office. The verification took place in the years 2019–2020 [28]

## 3. Results

The result of the calculation shows a statistically significant correlation between the volume of unplanned logging and the price of biomass, including paper pulp, wood

chips, and other biomaterials derived from wood production. The predictive model is in concurrence with economic theory, since the quantity of a product determines its price. The dependence is presented in the calculations of dependence before proposing an economic-mathematical model that allows for a prediction of the development of spruce biomass prices. The spruce biomass production in the Czech Republic is growing at a very fast pace so for practical reasons it is necessary to offer a possible tool for estimating the price of wood biomass in the short-term. The data used was provided by the Czech Statistical Office as well as companies operating in the wood and bio-energy industries. The data were verified, and the economic-mathematical prediction model was compiled.

Table 1 shows the data from which the result was calculated. The first column shows the time period (2010 to 2020), which is for a long enough duration for the results and forecasts to be valid. The data table shows the volume of unplanned salvage logging in thousands/m$^3$. Mining in 2020 was calculated using linear regression, as described in the methodology. The next column shows the price of spruce wood in EUR/m$^3$. The last column on the right shows the price of pulp in EUR/m$^3$. The price of both raw materials is also calculated for 2020 using linear regression. The price in EUR was always used in the calculations. For the conversion from CZK to EUR, the current exchange rate is used (19 March 2020) in the ratio of CZK 26.63 per EUR 1.

**Table 1.** Volume of the unplanned salvage logging and price of spruce wood (in EUR/m$^3$) between 2010 and 2021.

| Unplanned Logging 2010–2020 and the Forecast for the Year 2021 | Thousand m$^3$ ($X_i$) | Prices of Fuel Spruce Wood EUR/m$^3$ (Converted According to the Exchange Rate 26.63 CZK for EUR 1) | Prices of Spruce Pulp in EUR/m$^3$ (Converted at the Current Exchange Rate of CZK 26.63 for EUR 1) |
|---|---|---|---|
| 2010 | 3459 | 35 | 39 |
| 2011 | 3820 | 35 | 42 |
| 2012 | 3237 | 35 | 41 |
| 2013 | 4248 | 33 | 41 |
| 2014 | 4527 | 32 | 39 |
| 2015 | 8153 | 32 | 38 |
| 2016 | 9399 | 31 | 29 |
| 2017 | 11,743 | 30 | 30 |
| 2018 | 23,013 | 29 | 26 |
| 2019 | 25,000 | 23 | 23 |
| 2020 | 31,000 | 22 | 15 |
| 2021 | 36,000 | 21 | 13 |

Source: Own calculations, ČSU (Czech Statistical Agency) 2020, Forestry.

Table 2 shows the calculations of the descriptive statistics. It is especially interesting what is in the long-term average of unplanned logging over a period of ten years. The average harvest in the period from 2010 to 2019 was almost 15.5 million m$^3$ of spruce wood per year. If production increases to 31 million m$^3$ in 2020, unplanned production is expected to be 23 million m$^3$ higher than the long-term average. This is more than double the random extraction compared with the long-term average. This volume of accidental unplanned harvesting strongly affects not only the prices of bark, but also the prices of wood biomass, spruce wood, and spruce pulp for the manufacturing of paper, paper products, and paper packaging materials.

**Table 2.** Standard Deviation and Variance—Volume of the unplanned salvage logging.

| Volume of Unplanned Logging (2010–2020) | | |
|:---:|:---:|:---:|
| **Scores** | **Deviation (X − M)** | **Squared Dev.** |
| 3459 | −8140.91 | 66,274,400.83 |
| 3820 | −7779.91 | 60,526,985.46 |
| 3237 | −8362.91 | 69,938,248.46 |
| 4248 | −7351.91 | 54,050,567.28 |
| 4527 | −7072.91 | 50,026,043.01 |
| 8153 | −3446.91 | 11,881,182.28 |
| 9399 | −2200.91 | 4,844,000.83 |
| 11,743 | 143.09 | 20,475.01 |
| 23,013 | 11,413.09 | 130,258,644.10 |
| 25,000 | 13,400.09 | 179,562,436.37 |
| 31,000 | 19,400.09 | 376,363,527.28 |
| M: 11,599.91 | | SS: 1,003,746,510.91 |

Source: Own calculations, ČSU (Czech Statistical Agency) 2020, Forestry.

The results of the descriptive statistics (Years 2010–2020) in Tables 3 and 4 shows the average prices of wood and biomass products (firewood and spruce pulp for paper production). The table shows the calculation of the miscible deviation from average long-term prices. The average price of firewood was around 30 EUR/m$^3$ in 2019. In recent years, however, prices have been below this long-term average. Estimated prices for 2020 are expected to be almost 9 EUR/m$^3$ below the long-term average. The same applies to spruce pulp that is used for the production of paper and packaging materials from paper. The lowest average price of spruce pulp was 33 EUR/m$^3$ in 2019. Spruce pulp prices for paper production fell by EUR 10 below the long-term average. According to estimates for 2021, prices will continue to fall to the level of 13 EUR/m$^3$. This is a decrease of EUR 18 below the long-term average.

**Table 3.** Standard Deviation and Variance—The price of firewood.

| EUR/m$^3$ | Deviation (X − M) | Squared Dev. |
|:---:|:---:|:---:|
| 35 | 5.17 | 26.69 |
| 35 | 5.17 | 26.69 |
| 35 | 5.17 | 26.69 |
| 33 | 3.17 | 10.03 |
| 32 | 2.17 | 4.69 |
| 32 | 2.17 | 4.69 |
| 31 | 1.17 | 1.36 |
| 30 | 0.17 | 0.03 |
| 29 | −0.83 | 0.69 |
| 23 | −6.83 | 46.69 |
| 22 | −7.83 | 61.36 |
| 21 | −8.83 | 78.03 |
| M: 29.83 EUR/m$^3$ | | SS: 287.67 |

Source: Own calculations, ČSU (Czech Statistical Agency) 2020, Forestry.

**Table 4.** Standard Deviation and Variance—The Price of spruce wood pulp for industrial use.

| EUR/m$^3$ | Deviation (X − M) | Squared Dev. |
|:---:|:---:|:---:|
| 39 | 7.67 | 58.78 |
| 42 | 10.67 | 113.78 |
| 41 | 9.67 | 93.44 |
| 41 | 9.67 | 93.44 |
| 39 | 7.67 | 58.78 |
| 38 | 6.67 | 44.44 |
| 29 | −2.33 | 5.44 |
| 30 | −1.33 | 1.78 |
| 26 | −5.33 | 28.44 |
| 23 | −8.33 | 69.44 |
| 15 | −16.33 | 266.78 |
| 13 | −18.33 | 336.11 |
| M: 31.33 EUR/m$^3$ | | SS: 1170.6 |

Source: Own calculations, ČSU Czech Statistical Agency) 2020, Forestry.

Figure 1 shows the evolution and annual increments of unplanned spruce logging. The wood harvested was infested with bark beetles and is referred to as calamitous wood. In terms of wood quality (both physical and visual properties) it is wood that is comparable to the wood in series II (Figure 1) or the best-selling firewood in series III (Figure 2). The increase in calamitous wood has been drastic in the last five years and is unprecedented for Czech forestry. It is estimated that the bark beetle calamity in 2008 caused more than EUR 1.5 billion in damage to the Czech economy.

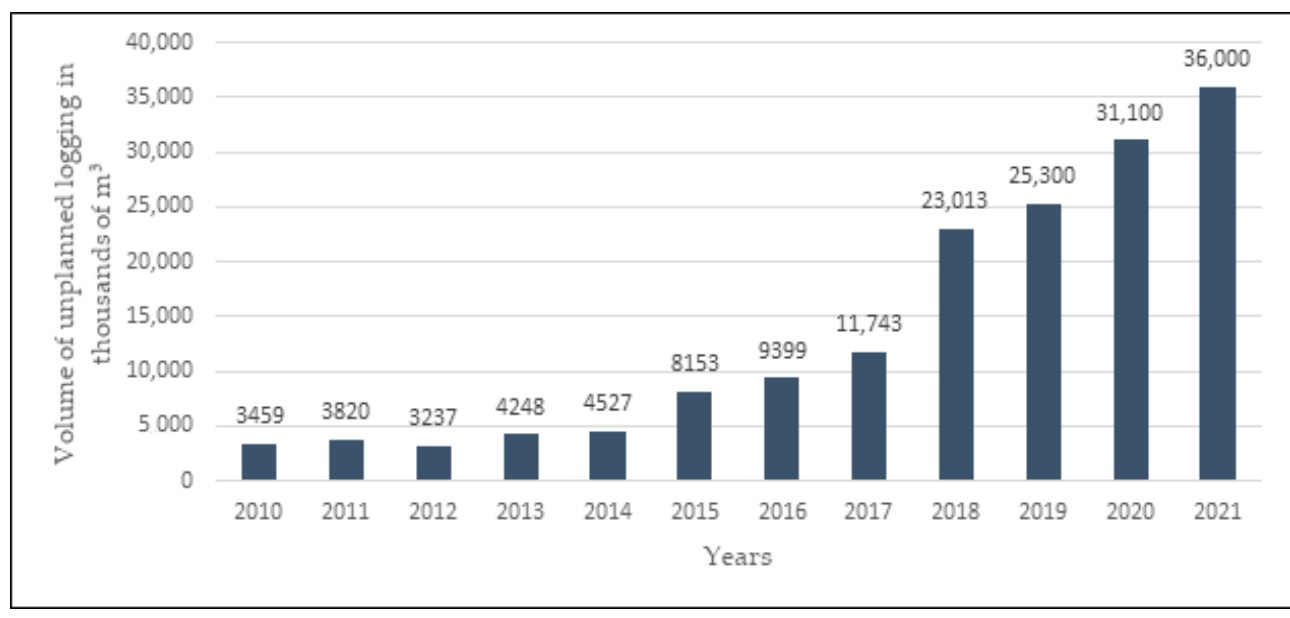

**Figure 1.** Volume of unplanned logging in the Czech Republic (from 2010 to 2020 and forecast for 2021); Source: ČSU (Czech Statistical Agency) 2020, Forestry.

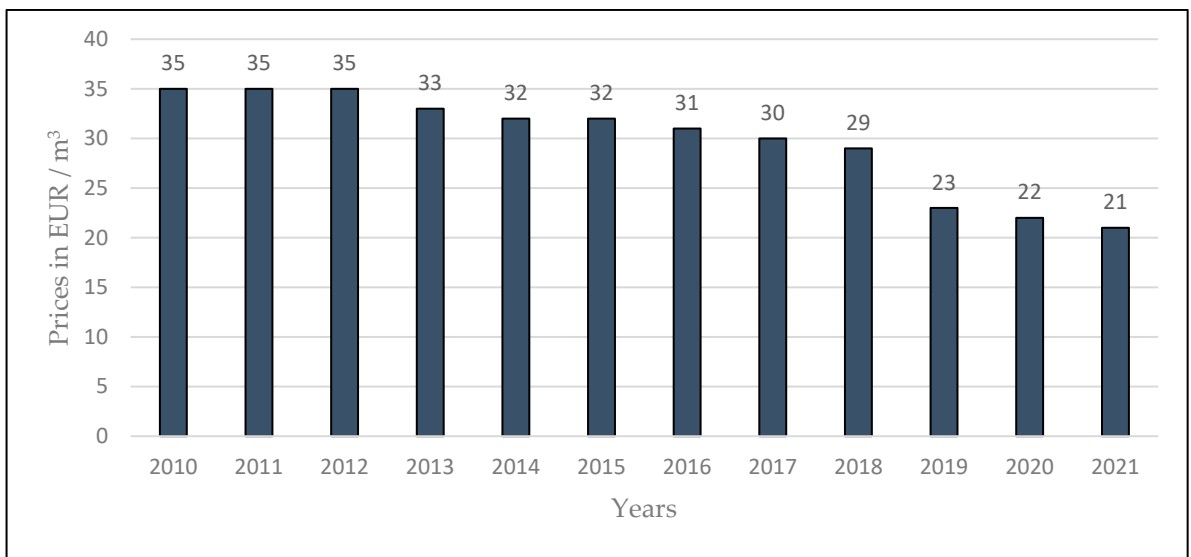

**Figure 2.** The Development of firewood prices and estimate for 2021 (Czech Republic); Source: ČSU (Czech Statistical Agency) 2020, Forestry.

Figure 2 shows firewood prices from 2010 to 2020, including a price forecast for 2021. Firewood is a commonly used secondary source of bioenergy for domestic heating. This method of energy use is traditional in Czech history and is still used very often today in the countryside. Wood that cannot be used in the construction and furniture industries is processed into firewood. A possible alternative use of waste wood and wood biomass is in the form of pellets and wood briquettes. These are increasingly sought-after items, and because the biomass product contains the added value of processing, their price is slightly higher than the price of traditional lumber. However, prices are still falling. The results of our long-term research show that the price of wood biomass for the production of firewood and wood pellets has decreased the most in the last five years. This indicates a strong increase in growth for the volume of unplanned logging. The decline in the price of wood biomass will cause economic problems, especially for producers of wood pellets and briquettes. Even today, the production costs of these organic products are often higher than the market price.

Figure 3 shows the rapid decline in wood pulp prices. The sharp decline was the largest in 2016. Over the years, pulp prices have decreased by almost 10 EUR/m$^3$. In the past five years, the price has fallen to the level of 24 EUR/m$^3$. The prediction for 2021 is EUR 13. However, the pulp market may further deteriorate since paper production will decline rapidly in the second quarter of 2020 due to the coronavirus crisis. At present (December 2020) no paper or packaging material is being produced and during the declared state of emergency, production was completely stopped in the Czech Republic.

Figure 4 shows that firewood prices will fall as the volume of unplanned logging increases. If the current trend is maintained and unplanned logging is increased to 36 million m$^3$ during 2021, then the price of firewood could reach 21 EUR/m$^3$. The model that was used for the calculation is y = −0.0004x + 35.489. The model is valid. $R^2$ has a value of almost 93%, which means that nine out of ten variables correspond to the model. In accordance with this model, we can also predict changes in the unit price depending on the further increase in unplanned spruce logging. The same linear predictive model was used for the calculation. If ceteris paribus increases unplanned logging by one million m$^3$ of spruce wood, the average price of firewood will decrease by EUR 2.

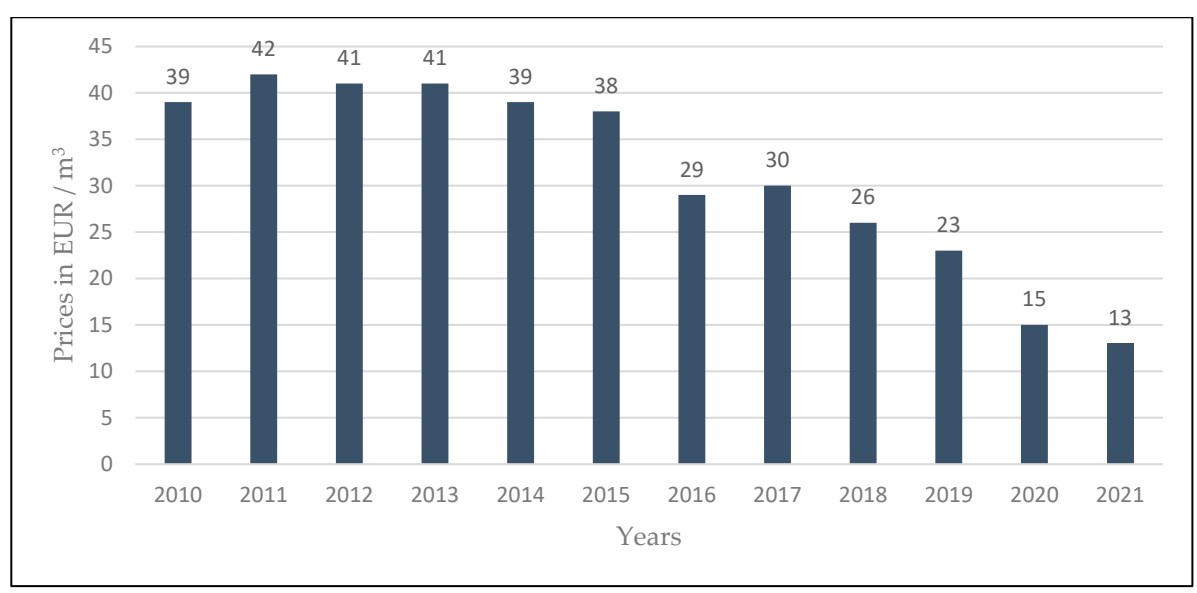

**Figure 3.** Price of pulp in the Czech Republic; Source: ČSU (Czech Statistical Agency) 2021, Forestry.

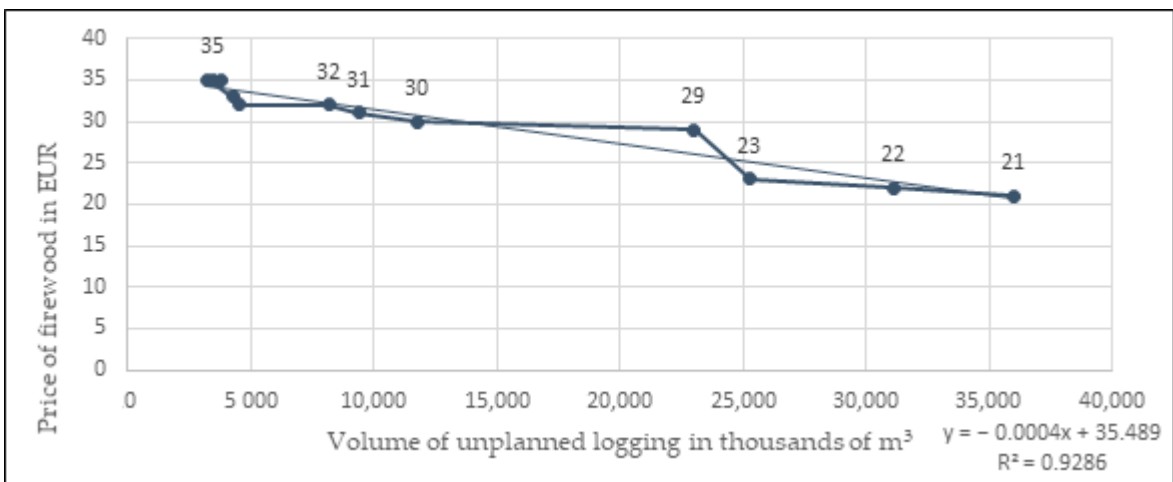

**Figure 4.** Dependence of the price of firewood on the volume of unplanned logging; Source: ČSU, (Czech Statistical Agency) 2020, Forestry.

Similarly, the calculation in Figure 5 is based on the relationship between the volume of unplanned logging and the price of spruce pulp. The predictive model allows for a relatively accurate estimate (y = −0 0008x + 42.815) of price developments. If unplanned logging reaches 36 million m³ of spruce wood, spruce pulp prices will fall to 13 EUR/m³. The mathematical model is 94% valid, which means that more than 90% of the values correspond to the model.

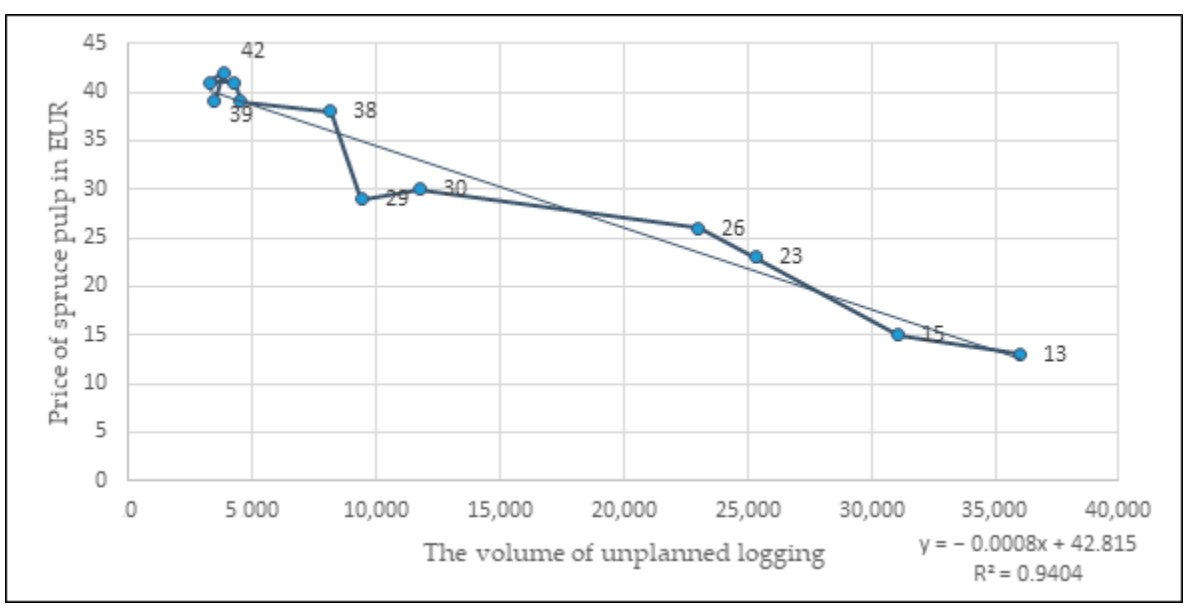

**Figure 5.** Dependence of the price of spruce pulp on the volume of unplanned logging; Source: ČSU; (Czech Statistical Agency) 2020, Forestry.

## 4. Discussion

The factors that influence the price of biomass can be divided into two basic areas. The price of biomass, which is intended for processing into wood pellets or wood briquettes for combustion, is determined by the quantity determined by normal market demand [29,30]. However, this economic factor little affects the price of biomass [31]. As some studies [31–33] show, demand has been rather constant in recent years [31,32]. Market demand is determined by both households [33] and enterprises [33,34]. However, households are gradually switching very slowly from the use of fossil fuels [35] and wood to gas or electricity-based thermal management [36], including the use of heat pumps. More than one million households in the Czech Republic use firewood, i.e., 23.3% of all households in the Czech Republic [7,37]. Households that use wood and, to a lesser extent, other biomass (straw and hay) for heat production have little impact on the price of wood biomass [38,39]. According to research, demand for firewood is stagnating, as there are more and more customers heating with firewood, but the amount of timber sold per customer decreases every year [39,40]. Ten years ago, vendors sold firewood on average 10 cubic meters of sprinkled wood or split wood per customer [39,40], when consumers used cheap but laborious heating. Year after year, this quantity is reduced per customer [39]. This is also because the population is aging and wood for heating is being used slowly less. Another reason why the average consumption of firewood per household is decreasing is that more and more customers are warming the family home, using subsidies for gas boilers [40,41]. Insulation reduces the consumption of firewood or wood fuel biomass by an average of 60% [39]. However, demand from companies processing wood biomass in the Czech Republic is growing faster. Wood biomass is used for the production of packaging materials, wood briquettes, and paper. In the Czech Republic, 882,122 tons of paper and paperboard were produced in 2020, up from 4.6% [39] year-on-year. According to data of the Czech Statistical Office, the number of economic operators in 2020 in the whole section CZ NACE 17 (production of paper and cellulose) totaled 979 business entities [39]. Of this number, 705 were working owners (natural businesspersons). The number 19,000 workers were employed throughout the section and the average wage was CZK 30,615 [42]. The development of sales of paper producers and, in particular, paper products is strongly linked to the development of the economy, as it also includes the area of paper packaging production [39,43,44] which is strongly linked to the retail trade. Strong sales growth of 5% continued in 2020 [39] despite the negative economic effects of the COVID pandemic.

Total sales of paper producers and paper products totaled CZK 83, 3 billion [39,44] in 2020. The added value of the whole section amounted to CZK 20 billion. The profit after the tax was CZK 6 billion [44–49]. The demand for wood biomass in the form of wood pulp will increase, but the price will not [48] due to the bark beetle disaster. Our results show a temporary contradiction between economic theory and the reality of Czech forestry.

Another economic factor affecting the price of wood biomass is bioenergetics demand [50–58] or, above all, technological capacity for the processing of so-called first and second-generation biofuels [14,59,60]. However, this problem goes beyond the objectives of our research. However, a fundamental factor in the development of the price of wood biomass is the biological factor, which is the amount of bark wood harvested in the Czech Republic [39]. This factor dominates. The average of mining, including planned and unplanned mining, was 15.5 million $m^3$ of wood by 2017. In 2020, only unplanned extraction of 35 million $m^3$ of contaminated spruce wood [39] was achieved. In 2021, it can even be 60 miles $m^3$ of the bark-wood. The bark wood can be efficiently processed by the crushing process [13]. Using crushing and drying technology [22,49], it creates a feedstock for processing wood briquettes and wood pellets. Wood sawdust resulting from the manufacture of wood products [61–65], planks, beams, etc., is processed similarly. Fuelwood products may improve the availability of heat supplies to households in the future. However, the problem remains that combustion releases $CO_2$ into the air [66,67], thus worsening the carbon balance. The more massive use of wood fuel products can cause negative environmental impacts [67,68]. Nevertheless, it can be argued that a market surplus of bark wood can be used temporarily. The fall in prices can thus benefit the manufacturing sector and domestic producers economically [69,70].

This paper has some shortcomings. First, our paper studies the impacts of calamity logging on the sustainable development of spruce fuel biomass prices and spruce pulp prices in the Czech Republic, but there are still some factors, such as altitude, minerals, human activities, and national policies, for which it is difficult to obtain data, or difficult to carry out quantitative analysis for other objective reasons, and which this paper did not consider. These factors directly or indirectly affect the sustainable development of spruce fuel biomass prices and spruce pulp prices, so more attention needs to be paid to them and more efforts made in future studies.

Additionally, a predictive model analysis was used to carry out the research in our paper. However, there are still some defects that need to be improved to carry out more comprehensive research. The predictive model analysis method is not fixed and is not a one-and-done prediction as it is validated and revised regularly to incorporate changes in the underlying data.

## 5. Conclusions and Recommendation

The current mission for global energy industries is the production of the cleanest possible energy. Developed countries strive to reduce industrial and carbon emissions, leveling the carbon footprint, and thus contributing to the protection of the air and the environment. In our research, we used empirical data on the market prices of these commodities, as well as data on the volume of unplanned mining. The result and conclusion of our calculations is an estimate of the price of wood biomass and wood pulp in 2021. The decrease in the price of wood biomass is not due to a decrease in demand, but to an increased volume of unplanned mining. This trend will continue this year and the following year. There is a need to support activities and support tools that create the conditions for processing surplus wood biomass at home and allow households to use this wood production for an affordable and renewable heat and energy source.

The results and calculations show that biomass (from spruce wood) is a promising raw energy material. Households can be the main customer. The assumption is that the price of wood biomass will remain stable. Likewise, wood pulp is economically promising as a raw material for paper production. Packaging paper material is a demanded raw material, especially at this time when transport to households requires an increased volume

of packaging use. Wood pulp is also a raw material for the production of hygienic and medical material. Together with the high production of calamitous wood and wood biomass, these production areas are becoming promising for investors. It is a raw material that is not only economically interesting but also sustainable and environmentally friendly. Our paper recommended some relevant policy implementations.

First, it is necessary to support the use of wood biomass systematically, strategically, and long-term; not only as a traditional building material and for furniture, but also in the production of paper packaging (which is environmentally friendly) and heat production (wood pellets and briquettes) for households.

Second, the problem in the Czech Republic is that logging is determined by the action of bark beetles. Therefore, logging is three times longer than the long-term average. Wood cannot be processed in such quantities at home (even as firewood) and sales abroad (including China) are severely limited due to the coronavirus pandemic. Therefore, instead of carbon sequestration by Czech forests, the opposite effect will occur.

Third, from the point of view of climate protection, the use of first or second-generation biofuels does not make a significant difference. The carbon dioxide produced when painting biomass products is recycled between the atmosphere and plants in the long run. In this sense, it does not matter whether it is the first fuel of the second or second generation, especially in comparison with the release of carbon from fossil fuels. Rather, there is a need today to treat bio-methane leakage from the putrefaction processes of organic materials in biogas production. This risk also applies to the natural rotting processes of improperly stored harvested bark wood, which remains in the forest or at the edge of the forest.

Lastly, the European Commission must draw up a strategic plan for the implementation of sustainable food and agriculture systems, forestry, and the production of products from biological materials. It must also set up an instrument to support EU bioeconomic policies under Horizon 2020. This will help EU countries prepare national and regional bioeconomy programs. The Commission must also develop the bioeconomy in rural, coastal, and urban areas, such as in waste management or carbon sequestration. Additionally, the Commission must develop the timber trade as a tool for the efficient use of surplus wood from unplanned logging energy.

**Author Contributions:** Conceptualization, K.M. (Karel Malec); M.M.; and D.T.; methodology, K.M. (Karel Malec); J.J.; and R.R.; software, S.N.K.A.-K.; P.P.; K.M. (Kamil Maitah); validation, K.M. (Karel Malec); M.M.; D.P.; D.T. and J.J.; formal analysis, K.M. (Karel Malec); S.N.K.A.-K. and M.M.; investigation, S.N.K.A.-K.; M.M.; K.M. (Karel Malec); and resources, K.M. (Karel Malec); P.P.; D.P.; and J.J; data curation, K.M. (Karel Malec); K.M. (Kamil Maitah); S.N.K.A.-K.; and J.J.; writing—original draft preparation, K.M. (Karel Malec); D.T.; M.M.; P.P.; and S.N.K.A.-K.; writing—review and editing, S.N.K.A.-K.; K.M. (Karel Malec); M.M.; and R.R.; visualization, K.M. (Karel Malec); M.M.; K.M. (Kamil Maitah); and D.P.; supervision, M.M.; P.P.; and J.J.; project administration, K.M. (Karel Malec); S.N.K.A.-K.; and J.J.; funding acquisition, K.M. (Karel Malec); M.M.; and D.T. All authors have read and agreed to the published version of the manuscript.

**Funding:** This paper was supported by the Internal grant agency (IGA) of the Faculty of Economics and Management, Czech University of Life Sciences Prague, grant no. 2019B0011 "Economic analysis of water balance of the current agricultural commodities production mix in the Czech Republic" (Ekonomická analýza vodní bilance stávajícího produkčního mixu zemědělských komodit v ČR).

**Acknowledgments:** We would like to thank the Internal grant agency (IGA) of the Faculty of Economics and Man-agement, Czech University of Life Sciences Prague for providing support and funding for this project.

**Conflicts of Interest:** The authors declare no conflict of interest.

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
