# Peer review of "The Impacts of Calamity Logging on the Sustainable Development of Spruce Fuel Biomass Prices and Spruce Pulp Prices in the Czech Republic"

_forests, doi:10.3390/f13010097_

Round 1
Reviewer 1 Report
Dear Authors,
The paper is in accordance with my field of research and an interesting topic and I therefore accepted to review this manuscript. The aim of the research is fairly interesting accompanied by an appropriate study methodology. However, the inroduction section is too long (4 pages). The readers need more information on bark beetles (biology, action, damage…etc). The paper contains some sloppy/careless mistakes. Also, it suffers from a narrow perspective (discussion section) and a need for reorganization of some parts. Thus, I judge that the article needs a major revision before publishing.
Please consider all the comments and questions in attached file for the revision of your manuscript.
Kindly

Author Response
Response to Editor and Reviewers
First, we would like to express our gratitude for the work you dedicated for investigating our research, identifying the points for improvement, and suggesting ways for achieving that. We are fully aware that your suggestions and recommendations are very important for improving our research and the way it is presented in this article.
This letter is to confirm that all final corrections and suggestions from the Editor-in-Chief and Reviewers have been addressed, corrected, and implemented. In addition, the authors have reviewed the manuscript thoroughly for possible style and language changes or substitutions.
Thank you and we hope we have answered all your suggestions and recommendations and improved our research.
Reviewer’s Comments:
“Dear Authors,
The paper is in accordance with my field of research and an interesting topic, and I therefore accepted to review this manuscript. The aim of the research is fairly interesting accompanied by an appropriate study methodology. However, the introduction section is too long (4 pages). The readers need more information on bark beetles (biology, action, damage…etc.). The paper contains some sloppy/careless mistakes. Also, it suffers from a narrow perspective (discussion section) and a need for reorganization of some parts. Thus, I judge that the article needs a major revision before publishing. Please consider all the comments and questions in attached file for the revision of your manuscript.”
Please find the following points which have been addressed.
- This sentence has been added:
The bark beetle calamity has a long-term impact on wood price formation, as the calamity lasts for several consecutive seasons and unplanned logging continues. In the Czech Republic, the above-average extent of unplanned mining has continued since 2017. In contrast, other abiotic causes of unplanned mining, such as wind, are usually resolved during the calamity season. The impact on wood pricing is rather short-lived.
- A literary reference has been added to this sentence:
Winkel [9] states that forest bio-economy is an economic activity involving forests and forest ecosystem services, including biomass-based value chains and the economic exploitation of other forest ecosystem resources energy [7,8].
- A literary reference has been added to this sentence:
This is linked to the promotion of sustainable forestry in other EU countries such as Finland, Sweden, Germany, and Austria, which have transformed the demand and production of timber. In addition, these wood-producing countries coordinate their activities energy [8]
- A literary reference has been added to this sentence:
However, the consumption of newsprint and graphic paper is expected to decrease [15-17].
- A literary reference has been added to this sentence:
In the Czech Republic, craft pulp is currently produced in paper mills in ŠtÄ›tí (Czech Republic) energy [7, 20]. The pulp produced is, therefore, free of cooking chemicals, bleach, and soil. The pulp obtained is mechanically modified according to the type of paper produced [20].
- A literary reference has been added to this sentence:
Wood biomass is used as a suitable and renewable source. So far in the European Union, only 60 percent of home-produced wood biomass is processed in green energy [14, 16].
- A literary reference has been added to this sentence:
Additionally, the Commission must develop the timber trade as a tool for the efficient use of surplus wood from unplanned logging energy [14-16].
In a nutshell, there have been a proper revision of the manuscript thanks to all the comments and recommendations of the reviewers.

Reviewer 2 Report
Dear Authors,
Article and topic has a very big potential, however the Introduction need to be very deep review. It would be very good to present your results and approach in much wider perspective and back ground. It would be very good to present how the price level is effected by higher, unplanned wood harvest in wider perspective. Especially in aspect large-scale natural disasters such as windbreaks, insect outbreaks, snow damage, droughts in other EU countries. In last decade we have in Europe several big scale forest damages in Sweden, Germany, Austria, France, Poland, which increase harvesting operations very much. How the higher wood harvesting quantity effect wood and biomass price level? How it is related with ownership of the forest? It is the same price decrease in state own forest and private?
Detail points:
Pulp production process is really needed? For what propose?
Several references needed. For example in lines 55, 56, 57, 75, 76, 110, 113, 193, 204.
Line 60 – this sentence is unclear.
Line 75, 76 – Which experts?
Line 113 - Forest cover please give the reference. You can find several Europe reports where forest cover for all countries are described.
Figure 1 – for better understanding unity should be changed from thousand in to millions.
Best regards,
Reviewer.
Author Response
Response to Editor and Reviewers
First, we would like to express our gratitude for the work you dedicated for investigating our research, identifying the points for improvement, and suggesting ways for achieving that. We are fully aware that your suggestions and recommendations are very important for improving our research and the way it is presented in this article.
This letter is to confirm that all final corrections and suggestions from the Editor-in-Chief and Reviewers have been addressed, corrected, and implemented. In addition, the authors have reviewed the manuscript thoroughly for possible style and language changes or substitutions.
Thank you and we hope we have answered all your suggestions and recommendations and improved our research.
Reviewer’s comments:
“Dear Authors,
Article and topic has a very big potential, however the Introduction need to be very deep review. It would be very good to present your results and approach in much wider perspective and back ground. It would be very good to present how the price level is effected by higher, unplanned wood harvest in wider perspective. Especially in aspect large-scale natural disasters such as windbreaks, insect outbreaks, snow damage, droughts in other EU countries. In last decade we have in Europe several big scale forest damages in Sweden, Germany, Austria, France, Poland, which increase harvesting operations very much. How the higher wood harvesting quantity effect wood and biomass price level? How it is related with ownership of the forest? It is the same price decrease in state own forest and private?
Detail points:
Pulp production process is really needed? For what propose?
Several references needed. For example in lines 55, 56, 57, 75, 76, 110, 113, 193, 204.
Line 60 – this sentence is unclear.
Line 75, 76 – Which experts?
Line 113 - Forest cover please give the reference. You can find several Europe reports where forest cover for all countries are described.
Figure 1 – for better understanding unity should be changed from thousand in to millions.
Best regards,
Reviewer.”
Please find the following points which have been addressed.
- This sentence has been added:
The bark beetle calamity has a long-term impact on wood price formation, as the calamity lasts for several consecutive seasons and unplanned logging continues. In the Czech Republic, the above-average extent of unplanned mining has continued since 2017. In contrast, other abiotic causes of unplanned mining, such as wind, are usually resolved during the calamity season. The impact on wood pricing is rather short-lived.
- A literary reference has been added to this sentence:
Winkel [9] states that forest bio-economy is an economic activity involving forests and forest ecosystem services, including biomass-based value chains and the economic exploitation of other forest ecosystem resources energy [7,8].
- A literary reference has been added to this sentence:
This is linked to the promotion of sustainable forestry in other EU countries such as Finland, Sweden, Germany, and Austria, which have transformed the demand and production of timber. In addition, these wood-producing countries coordinate their activities energy [8]
- A literary reference has been added to this sentence:
However, the consumption of newsprint and graphic paper is expected to decrease [15-17].
- A literary reference has been added to this sentence:
In the Czech Republic, craft pulp is currently produced in paper mills in ŠtÄ›tí (Czech Republic) energy [7, 20]. The pulp produced is, therefore, free of cooking chemicals, bleach, and soil. The pulp obtained is mechanically modified according to the type of paper produced [20].
- A literary reference has been added to this sentence:
Wood biomass is used as a suitable and renewable source. So far in the European Union, only 60 percent of home-produced wood biomass is processed in green energy [14, 16].
- A literary reference has been added to this sentence:
Additionally, the Commission must develop the timber trade as a tool for the efficient use of surplus wood from unplanned logging energy [14-16].
In a nutshell, there have been a proper revision of the manuscript thanks to all the comments and recommendations of the reviewers.

Round 2
Reviewer 1 Report
Dear authors,
Most of my comments (90%) were not taken into account during the revision ! how is that possible?
Comments that were not taken into account are :
- What type of forests (High forests, CWS, coppice)? species?
- Move to the end of Intro lines 21-23 in the second version
- Prices fall dramatically ! give an order of magnitude or a percentage
- Reference? more than 42% according to eurostat
- Round please (2%) or 2.1 use max 2 decimals
- Not okey please remove this statement "suggesting that the role of the forestry sector is more important than agriculture in the Czech Republic"
- do not use 3 decimals it has no sens
- Colors not Necessary, use hashed bar for 2021
- And reduce many pragraphs according to my comments on the first manuscript ...
Please find attached my comments (same as the first version), send a point-by-point response.
Kindly

Author Response
Response to Editor and Reviewers
We would like to thank the Reviewer for their interest in our work and for their helpful comments that have greatly improved the manuscript. We did our best to respond to the points raised. The Reviewer brought up some good points, and we appreciate the opportunity to clarify our research objectives and results.
As indicated below, we considered all the concerns and specific comments provided by the Reviewer and made necessary changes accordingly.
We have changed the article due to your recommendation. And we also checked and changed the English by a native speaker and style of the article.
Thank you for the opportunity to review this interesting article. However, we have made the following findings as follows:
- Line number 55. This sentence has been added.
In the Czech Republic, coniferous stands make up 71.5%, of which Norway spruce accounts for 50.0%. Spruce lichen and glossy lichen eater attacks just and most spruces. Furthermore, fir is 1.2%, pine is 16.2%, and larch 3.8%. Other coniferous forests represent 0.3%. Deciduous forests make up 27.3% of the forest area in the Czech Republic. Oak makes up 7.3%, beech 8.6%, birch 2.8%, other deciduous forests are 8.7%. The rest up to 100% are mainly temporary clearings, which will be afforested in the coming years or are intended for afforestation.
- Line number 67-69
The sentence was moved to the end of the introduction, as requested by the reviewer.
- Line number 84
Wood prices fell by 61% year on year.
- Line number 96
Two dots have been deleted.
- Line number 111
In the Czech Republic, forests cover approximately 2.9 million ha, which is 33.7% of the total area of land
- Line number 217
The figure has been corrected to 2%
- Line number 118
In the sentence, the figure was corrected to: 1.7%
- Line number 118-119
The sentence has been deleted.
- Line number 124-131
¨The sentence has been removed.
- Line number 139
Pulp and paper industry.
- Line number 145
Link number 21 has been added.
- Line number 215 -222
The paragraph has been deleted.
- Line number 226
The sentence has been deleted.
- Line number 230-246
The paragraph has been moved to the discussion.
Thank you for your comment.

Round 3
Reviewer 1 Report
Dear authors,
The introduction section still too long (4 pages ! too much), plase be brief and concise.
The use of colors in the figures does not make sense, consider using (gray and black) for printing in black only.
During the revision stage, you added a new co-author to the list of authors ! this is not ethical. All authors should approve the manuscript and agree before its submission. Could you please give more detail about his contribution ?
Kindly
Author Response
Response to Editor and Reviewers
We would like to thank the Reviewer for their interest in our work and for their helpful comments that have greatly improved the manuscript. We did our best to respond to the points raised. The Reviewer brought up some good points, and we appreciate the opportunity to clarify our research objectives and results.
As indicated below, we considered all the concerns and specific comments provided by the Reviewer and made necessary changes accordingly.
We have changed the article due your recommendation. And we also checked and changed the English by a native speaker and style of the article.
Thank you for the opportunity to review this interesting article. However, we have made the following findings as follows:
“Dear authors,
The introduction section is still too long (4 pages! too much), please be brief and concise.
The use of colors in the figures does not make sense, consider using (gray and black) for printing in black only.
During the revision stage, you added a new co-author to the list of authors! this is not ethical. All authors should approve the manuscript and agree before its submission. Could you please give more detail about his contribution?”
- The Introduction part has been adjusted to be brief and concise as recommended by the Reviewer.
- The use of colors for all the figures has been adjusted by using gray color as suggested by the Reviewer.
- Regarding the additional co-author, all the authors approved that, signed, and uploaded the “Authorship Change Form” of the Journal during the submission of the revised manuscript. Kindly note that the Editor of this Journal was well informed during the addition of the co-author. The co-author was added because he assisted us immensely (like writing, review, and editing) during the major revisions of the manuscript process.
Once again, thank you for your comments.
